



# Compensatory effects conceal large uncertainties in the modelled processes behind the ENSO-CO$_2$ relationship

István Dunkl[1,2], Ana Bastos[3], and Tatiana Ilyina[4,5,1]

[1]Max Planck Institute for Meteorology, Hamburg, Germany
[2]Institute for Meteorology, Leipzig University, Leipzig, Germany
[3]Max Planck Institute for Biogeochemistry, Jena, Germany
[4]CEN, Universität Hamburg, Hamburg, Germany
[5]Helmholtz-Zentrum Hereon, Germany

**Correspondence:** István Dunkl (istvan.dunkl@uni-leipzig.de)

**Abstract.** A large fraction of the interannual variations in the global carbon cycle can be explained and predicted by the impact of El Niño Southern Oscillation (ENSO) on net biome production (NBP). It is therefore crucial that the relationship between ENSO and NBP is correctly represented in Earth system model (ESMs). With this work, we look beyond the top-down ENSO-CO$_2$ relationship in 22 CMIP6 ESMs by describing their characteristic ENSO-NBP pathways. These pathways result from the

configuration of three interacting processes which contribute to the overall ENSO-CO$_2$ relationship: ENSO-strength, ENSO-induced climate anomalies, and the sensitivity of NBP to climate. The analysed ESMs agree on the direction of the sensitivity of global NBP to ENSO, but have very large uncertainty in its magnitude, with a global NBP anomaly of -0.15 PgC yr$^{-1}$ to -2.13 PgC yr$^{-1}$ per standardised El Niño event. The largest source of uncertainty is the differences in the sensitivity of NBP to climate. The uncertainty among the ESMs grows even further when only the differences in NBP sensitivity to climate are

considered. This is because differences in the climate sensitivity of NBP are partially compensated by ENSO strength. There is a similar phenomenon regarding the distribution of ENSO-induced climate anomalies. We show that even model that agree on global NBP anomalies have strong disagreements in the contribution of different regions to the global anomaly. This analysis shows, that while ESMs can have a comparable ENSO-induced CO$_2$ anomaly, the carbon fluxes contributing to this anomaly originate from different regions and are caused by different drivers. The consequence of these alternative ENSO-NBP pathways

can be a false confidence in the reproduction of CO$_2$ by assimilating the ocean, and the dismissal of predictive performance offered through ENSO. We suggest to improve the underlying processes by using large-scale carbon flux data for model tuning in order to capture the ENSO-induced NBP anomaly patterns. The increasing availability of carbon flux data from atmospheric inversions and remote sensing products makes this a tangible goal and would lead to a better representation of the processes driving the interannual variability of the global carbon cycle.

# 1 Introduction

The relationship between El Niño Southern Oscillation (ENSO) and atmospheric CO$_2$ observations at Mauna Loa was first reported by Bacastow (1976). Altered atmospheric circulation patterns during El Niño events cause warm and dry conditions



across the tropics, leading to a reduction in net biome production (NBP) due to reduced net primary productivity (NPP) and increased or decreased heterotrophic respiration (Rh) (Qian et al., 2008; Bastos et al., 2018). The ENSO-induced climate anomalies have a significant impact on the gross primary productivity (GPP) of 32% of the vegetated land area and can explain up to 26% of the interannual variation in global GPP (Zhang et al., 2019). Some El Niño events can be severe enough to turn the Amazon Basin, a carbon sink of global importance, into a net carbon source (Tian et al., 1998).

But ENSO does not only explain a large fraction of the NBP variability, it is also the main source of seasonal to decadal predictability in the earth system (Manzanas et al., 2014; Zeng et al., 2008; Spring and Ilyina, 2020; Li et al., 2022). Tropical carbon flux anomalies lag behind ENSO by three to six months (Zhu et al., 2017), meaning that even without further knowledge on the evolution of ENSO, near-term carbon flux variability can be anticipated based on the present ENSO conditions. On top of this lag effect, Earth system model (ESM) simulations starting in winter can predict ENSO conditions for up to one year (Barnston et al., 2019).

Further predictability is added to the system by the land surface, which prolongs the ENSO-induced climate anomalies. The larger the anomaly, the longer it will take for soil moisture and conditions to return to normality, and ENSO years are often among the most extreme years of variability (Holmgren et al., 2001). Even longer predictability mechanisms might be triggered through vegetation dynamics (Holmgren et al., 2001). This can happen in dry years through the lasting impact of defoliation and tree mortality (Wigneron et al., 2020; Santos et al., 2018), or through wildfire, which requires decades of recovery (Silva et al., 2018). Wet events, on the other hand, can provide long-term predictability as these events drive plant recruitment in semi-arid ecosystems (Holmgren et al., 2001). Extreme events play a crucial role in the vegetation dynamics of these ecosystems, where the establishment of trees and shrubs needs sustained wet conditions (Chang-Yang et al., 2016; López et al., 2006).

Because ENSO plays such a large role in the variability and predictability of NBP, the correct representation of the related processes in ESMs is especially important. Three key processes that explain the relationship between ENSO and NBP can be arranged hierarchically. On the highest level of this hierarchy is the strength of the ENSO events. Despite considerable advancements in our understanding of ENSO dynamics, there remains a wide range of simulated ENSO amplitude in ESMs (Beobide-Arsuaga et al., 2021). The amplitude of ENSO can be measured as the standard deviation (SD) of sea surface temperature anomalies (SSTA) in the Niño3.4 region (170° W–120° W, 5° N–5° S), and ranges between 0.4 °C and 1.4 °C in models from the Coupled Model Intercomparison Project Phase 6 (CMIP6) (Brown et al., 2020; Beobide-Arsuaga et al., 2021; Cai et al., 2022).

The second process that explains the impact of ENSO on NBP are the ENSO-induced climate anomalies. Most of the ENSO teleconnections are caused by a reorganisation of tropical convection patterns (Perry et al., 2020). These create temperature and precipitation anomalies in northern South America, Southeast Asia and northern Australia. However, the ENSO-induced changes in the upper atmosphere can create Rossby waves that propagate polewards and lead to climate anomalies in the mid-latitudes. The ENSO teleconnection strengths show a high uncertainty among CMIP5 models, with an average correlation to observed teleconnection patterns of 0.7 for temperature, and 0.46 for precipitation (Perry et al., 2020). Although the representation of the relationship between ENSO and tropical precipitation has improved from CMIP5 to CMIP6, there are still





considerable deviations from the observed relationship (Yang and Huang, 2022). The impact of ENSO on East Asian summer rainfall, for example, can only be captured by 11 out of 20 CMIP6 models (Fu et al., 2021).

At the last stage within the hierarchy of processes that shape the ENSO-NBP relationship are the differences in biogeochemistry, specifically, the sensitivity of NBP to local climatic anomalies caused by ENSO. Because of limitations in carbon flux observations and the covariability of climatic conditions, the contribution of temperature and moisture in driving the carbon cycle remains a debated topic in the literature (Piao et al., 2020). This uncertainty can be demonstrated by the sensitivity of atmospheric $CO_2$ growth rate to tropical temperature from reanalysis data, where determined sensitivities differ by a factor of two (Piao et al., 2020). Regional differences in the sensitivity of carbon fluxes to climate depend on ecosystem type and climate. Semi-arid ecosystems and tropical forests which cover most of the land area affected by ENSO, show the highest sensitivity to climate variability (Bastos et al., 2013; Poulter et al., 2014; Ahlström et al., 2015; O'Sullivan et al., 2020). However, these biomes are also where carbon flux sensitivities have the highest inter-model spread and bias to observations (O'Sullivan et al., 2020; Koirala et al., 2022).

The aim of this study is to look beyond the top-down relationship between ENSO and atmospheric $CO_2$ growth rate in ESMs and reveal the sources of uncertainty in the ENSO-NBP relationship. We quantify the specific ENSO-NBP pathways that describe the location and drivers of the ENSO-induced NBP anomalies. These pathways are characterised by three main processes that shape the ENSO-NBP relationship: ENSO strength, ENSO-induced climate anomalies, and biogeochemistry. We quantify how much the uncertainties in these three processes contribute to the uncertainty in the ENSO-NBP relationship among the ESMs and compare the ESMs to observations.

## 2 Methods

### 2.1 Data

We measure the interactions between ENSO, temperature, precipitation, and NBP in 22 CMIP6 ESMs and compare them with observation based data sources. In order to have a large sample size of ENSO conditions, we use unforced piControl simulations. The analysed ESMs, their variant labels, and simulation lengths are listed in Table 1. We decompose the NBP anomalies into the components

$$NBP \approx NPP - Rh - fire. \tag{1}$$

Because some ESMs simulate other types of disturbances than fires, (1) does not lead to an exact reproduction of NBP, but gives an approximate contribution of the large carbon fluxes to the land-air $CO_2$ exchange. Fire emissions are only available for 13 of the analysed ESMs. Although CMCC-CM2-SR5 simulates fire emissions, the data is not available online. Instead, we calculate fire emissions for CMCC-CM2-SR5 by inverting the mass balance in (1).

The carbon fluxes of the ESMs are compared with three additional data sets, which are called observations here. These observations are land-atmosphere carbon fluxes from the atmospheric inversion product Copernicus Atmosphere Monitoring Service (CAMS) (Chevallier et al., 2005), net ecosystem exchange (NEE) from upscaled flux tower measurements (FLUXCOM



**Table 1.** The Earth system models used in this study, as well as the experiment runs, simulations times and whether or not fire emissions are provided.

| ESM | Variant label | Years | Fire emissions | Reference |
|---|---|---|---|---|
| ACCESS-ESM1-5 | r1i1p1f1 | 1000 | | Ziehn et al. (2020) |
| AWI-ESM-1-1-LR | r1i1p1f1 | 100 | X | Shi et al. (2020) |
| BCC-CSM2-MR | r1i1p1f1 | 1374 | | Wu et al. (2019) |
| CanESM5 | r1i1p1f1 | 1400 | | Swart et al. (2019) |
| CESM2 | r1i1p1f1 | 1200 | X | Danabasoglu et al. (2020) |
| CMCC-CM2-SR5 | r1i1p1f1 | 500 | X | Lovato et al. (2022) |
| CMCC-ESM2 | r1i1p1f1 | 500 | X | Cherchi et al. (2019) |
| CNRM-ESM2-1 | r1i1p1f2 | 500 | X | Séférian et al. (2019) |
| E3SM-1-1 | r1i1p1f1 | 165 | X | Golaz et al. (2019) |
| EC-Earth3-CC | r1i1p1f1 | 1505 | X | Döscher et al. (2022) |
| GFDL-ESM4 | r1i1p1f1 | 500 | X | Dunne et al. (2020) |
| GISS-E2-1-G | r1i1p3f1 | 165 | | Orbe et al. (2020) |
| INM-CM5-0 | r1i1p1f1 | 1201 | | Volodin et al. (2018) |
| IPSL-CM6A-LR | r1i1p1f1 | 2001 | | Boucher et al. (2020) |
| MIROC-ES2H | r1i1p4f2 | 420 | | Watanabe et al. (2021) |
| MIROC-ES2L | r1i1p1f2 | 500 | | Hajima et al. (2020) |
| MPI-ESM1-2-LR | r1i1p1f1 | 1000 | X | Mauritsen et al. (2019) |
| MRI-ESM2-0 | r1i2p1f1 | 251 | X | Yukimoto et al. (2019) |
| NorCPM1 | r1i1p1f1 | 500 | X | Bethke et al. (2021) |
| NorESM2-LM | r1i1p1f1 | 300 | X | Seland et al. (2020) |
| NorESM2-MM | r1i1p1f1 | 500 | X | Seland et al. (2020) |
| UKESM1-0-LL | r1i1p1f2 | 1880 | | Sellar et al. (2019) |

version: ANN+CRUNCEPv6, Jung et al. 2019), and NBP from the TRENDYv6 ensemble of land surface models (Sitch et al.,

2015). Although NEE contains land-atmosphere carbon fluxes like geological $CO_2$ release and lateral fluxes which are not part of NBP (Ciais et al., 2022), these fluxes only play a smaller part in the global carbon cycle and unlikely to significantly affect the sensitivity to ENSO (Canadell et al., 2021). We additionally consider sea surface temperature (SST) reanalysis data (HadISST, Rayner et al. 2003), and temperature and precipitation from the ERA-Interim reanalysis (Dee et al., 2011). To account for the uncertainty in reanalysis data other reanalysis products were added to the analysis in addition to ERA-Interim. These products

are temperature from the Climatic Research Unit gridded Time Series (CRU v4, Harris et al. 2020), JRA-55 (Kobayashi et al., 2015), the Modern-Era Retrospective Analysis for Research and Applications, Version 2 (MERRA-2, Gelaro et al. 2017), the NCEP/NCAR 40-Year Reanalysis Project (NCEP, (Kalnay et al., 1996)), and the bias-adjusted ERA5 reanalysis data (WFDE5, Cucchi et al. 2020), and precipitation from JRA-55, MERRA2, and NCEP.





## 2.2 Processing and analysis

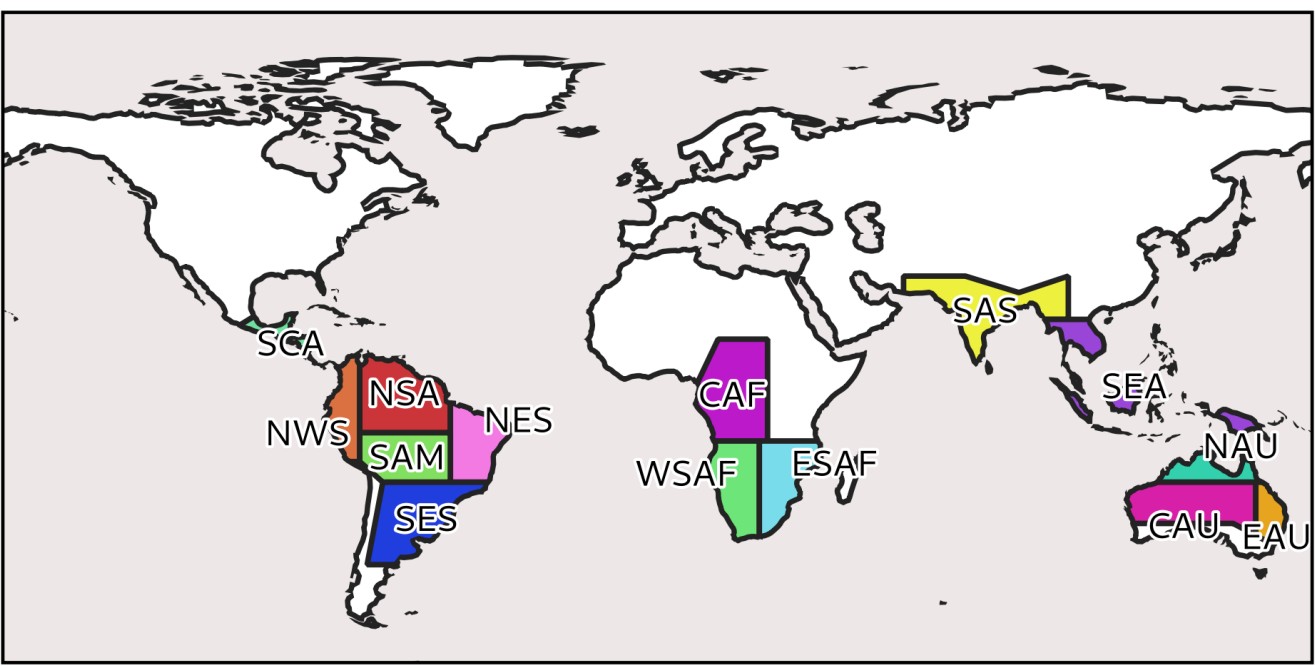

**Figure 1.** The 14 IPCC climate reference regions with the largest ENSO-induced NBP anomalies. S.E.Asia (SEA); N.South-America (NSA); N.E.South-America (NES); E.Southern-Africa (ESAF); South-American-Monsoon (SAM); N.Australia (NAU); N.W.South-America (NWS); Central-Africa (CAF); S.Asia (SAS); C.Australia (CAU); E.Australia (EAU); S.Central-America (SCA); W.Southern-Africa (WSAF); S.E.South-America (SES).

We calculate the annual anomalies of Niño3.4 SST, temperature, precipitation, and carbon fluxes for the analysis. In this study, we base annual averages on the time window from July to June next year, instead of January to December. We use this definition of years to better capture distinct ENSO events. ENSO SSTA usually peaks during boreal winter, and the warm El Niño events are often followed by a cold La Niña event (An and Kim, 2017). Years starting in January would therefore not be centred around event peaks and could contain the tail of an El Niño, and the beginning of a La Niña event. Annual anomalies of the data are obtained by subtracting the climatology from the ESM data, and by removing the linear trend from the observational products. ENSO strength is calculated as the SD of annual SSTA in the Niño3.4 region. NBP and climate anomalies are aggregated to regional averages using the boundaries of the updated IPCC reference regions (Iturbide et al., 2020). For this analysis we only focus on the regions with the strongest ENSO-NBP relationship. These regions are selected by averaging the ENSO-induced NBP anomalies over all data sources and selecting the 14 regions with the highest absolute ENSO-induced NBP anomalies (Fig. 1). We calculate the ENSO-induced climate and carbon anomalies as the coefficient of a





linear regression model with zero intercept:

$$\Delta Temp_{ijp} = \beta_{ET_{ij}} \times \Delta ENSO_{jp}, \qquad (2)$$

$$\Delta Precip_{ijp} = \beta_{EP_{ij}} \times \Delta ENSO_{jp}, \qquad (3)$$

$$\Delta NBP_{ijp} = \beta_{EN_{ij}} \times \Delta ENSO_{jp}, \qquad (4)$$

with $\Delta X_{ijp}$ as the annual anomalies, $\Delta ENSO_{jp}$ as the annual mean Niño3.4 SSTA, and the regression coefficients $\beta_{EX_{ij}}$ for region ($i = 1, \ldots, 46$, all IPCC regions with land surface), data source ($j = 1, \ldots, 25$, 22 ESMs and 3 observational NBP products using ERA-Interim climate), and year $p$. Although El Niño and La Niña do not produce entirely symmetrical responses in the atmosphere and the land system, we use this method for the ease of simplified results, assuming that Niño and Niña events are similar.

We describe the ENSO-NBP pathways by three distinct processes: ENSO-strength, ENSO-induced climate anomalies, and NBP sensitivity to climate (biogeochemistry). To quantify these pathways, we calculate the ENSO-induced NBP anomaly due to the differences in each of these processes respectively ($\Delta NBP^{ENSO}$, $\Delta NBP^{CLIM}$, $\Delta NBP^{BIO}$). This is done by considering only the differences in one of the processes at a time while using the mean conditions across all ESMs for the other processes. To compare the effects of similar ENSO events across ESMs, we calculate the 90$^{th}$ percentile of Niño3.4 SSTA for every ESM and HadISST. These standardised ENSO events are called ENSO-90 here and have a return interval of around 11

years. We calculate $\Delta NBP^{ENSO}$ by multiplying the mean global NBP sensitivity to ENSO ($\beta_{EN_j}$) with the ENSO-90 SSTA values of each model ($\Delta ENSO_{j90}$):

$$\Delta NBP_j^{ENSO} = \beta_{EN_j} \times \Delta ENSO_{j90}. \qquad (5)$$

The differences due to ENSO-induced climate anomalies, and NBP sensitivity to climate are assessed by fitting multiple linear

regression models of NBP ($MLR_{NBP}$) for each region and data source. These models predict NBP based on annual temperature and precipitation anomalies as:

$$\Delta NBP_{ijp} = \beta_{NT_{ij}} \times \Delta Temp_{ijp} + \beta_{NP_{ij}} \times \Delta Precip_{ijp}, \qquad (6)$$

with the $\Delta NBP_{ijp}$ as the NBP anomaly, and its sensitivity to temperature and precipitation as $\beta_{NT_{ij}}$ and $\beta_{NP_{ij}}$. To assess the differences due to ENSO-induced climate anomalies, we use $MLR_{NBP}$ with temperature and precipitation anomalies of

an ENSO-90 event for every ESM and averaged sensitivity values across all ESMs:

$$\Delta NBP_{ij}^{CLIM} = \frac{1}{22}\sum_{j=1}^{22}(\beta_{NT_{ij}}) \times \beta_{ET_{ij}} \times \Delta ENSO_{j90} + \frac{1}{22}\sum_{j=1}^{22}(\beta_{NP_{ij}}) \times \beta_{EP_{ij}} \times \Delta ENSO_{j90}. \qquad (7)$$

Vice versa, the differences due to biogeochemistry are calculated by using temperature and precipitation anomalies of an ENSO-90 event averaged across all ESMs with the model-specific NBP sensitivities to climate:

$$\Delta NBP_{ij}^{BIO} = \beta_{NT_{ij}} \times \frac{1}{22}\sum_{j=1}^{22}(\beta_{ET_{ij}} \times \Delta ENSO_{j90}) + \beta_{NP_{ij}} \times \frac{1}{22}\sum_{j=1}^{22}(\beta_{EP_{ij}} \times \Delta ENSO_{j90}). \qquad (8)$$





We compare the contribution of the three processes to the overall uncertainty by measuring the spread of the ENSO-NBP relationship among the ESMs as the coefficient of variation (CV (%); ratio of standard deviation to mean).

# 3 Results

## 3.1 Global ENSO-NBP relationship



**Figure 2.** The ENSO-induced NBP anomalies in ESM (green) and observational products (purple) for a) an El Niño event of a 90$^{th}$ percentile intensity (CV=48%, 0.39 without GFDL-ESM4), and the contribution of three processes to the differences in the ENSO-NBP relationship: b) ENSO strength (CV=32%), c) differences in ENSO-induced climate anomaly patterns (teleconnections, CV=27%), and d) differences in biogeochemistry (CV=56%).

The NBP anomalies of a standardised ENSO-90 event range between -0.15 and -2.13 PgC yr$^{-1}$, with a CV of 48%, which
we use as the reference value for the uncertainty in the ENSO-NBP relationship (Fig. 2 a). The mean ESM ENSO-90 NBP





**Figure 3.** The regional ENSO-induced NBP anomalies in ESM and observational products from an El Niño event of a 90th percentile intensity.

anomaly is -0.88 PgC yr$^{-1}$, which is between CAMS and TRENDY. The mean ENSO-90 NBP anomaly of the observations is dragged down by FLUXCOM, which is reported to underestimate the interannual variability of carbon fluxes (Jung et al., 2019). There is a strong disagreement on the regional contribution to global NBP anomalies (Fig. 3). This disagreement can be exemplified by SEA and NSA, the two regions contributing most to global NBP anomalies. The combined NBP anomalies of these two regions explain 48% of the global NBP anomalies across all ESMs. However, the global contribution of SEA and NSA ranges between -23% in MIROC-ES2H, which, unlike the other ESMs has positive NBP anomalies in SEA during El Niño, and NorESM2-LM, where SEA and NSA explain 70% of the global anomaly. There is also little agreement on the ratio



of SEA to NSA anomalies. Although the mean NBP anomaly from SEA is 21% larger than the mean anomaly from NSA, half of the 22 ESMs have larger anomalies in NSA than SEA.

Although the method used here does not take the asymmetry of ENSO events into account, we found this effect to be negligible on the global scale for most ESMs (Fig. A1).

## 3.2    ENSO-strength

ENSO strength varies between 0.35 °C (INM-CM5-0) and 1.39 °C (CMCC-ESM2), while the ESM mean (0.86 °C) is slightly higher than the HadISST reanalysis (0.76 °C, Fig. 4). Despite the wide range of ENSO strength, these differences are partially

offset by the sensitivity of global NBP to ENSO. ESMs with a strong ENSO tend to have a lower NBP sensitivity to Niño3.4 SSTAs.

To single out the effect of the differences in ENSO strength, we multiply the ENSO-90 SSTAs with the mean NBP sensitivity of -0.86 PgC yr$^{-1}$ °C$^{-1}$ SSTA. The resulting ENSO-induced NBP anomalies range between -0.39 and -1.64 PgC yr$^{-1}$. Considering only the differences in ENSO strength leads to CV of 32% which is a 33% reduction compared to the overall

ENSO-NBP CV (Fig. 2 b).

## 3.3    ENSO-induced climate anomalies

We compare the regional patterns of ENSO-induced climate anomalies between ESM and observations and assess how these differences affect the ENSO-NBP relationship. The ESMs generally capture the sign and strength of ENSO-induced temperature and precipitation anomalies (Figs. A2, A3). There are, however, regional differences in the uncertainty of climate

anomalies among the ESMs. The spread in ENSO-induced temperature anomalies is especially high in CAU, NAU and NSA, and there are high uncertainties in the ENSO-induced precipitation anomalies in SEA and NWS.

We use a statistical model with uniform NBP sensitivities to determine the effect of differences in ENSO-induced climate anomaly patterns on NBP (Fig. 2 c) and A4). Differences in ENSO-induced climate anomalies contribute less to the uncertainty in the ENSO-NBP relationship than the other two processes (CV=27%). Most of the global NBP anomalies fall within a similar

range, except for MPI-ESM1-2-LR. The above-average NBP anomalies of MPI-ESM1-2-LR are caused by the strong ENSO-induced temperature anomalies in several regions (Fig. A2).

The comparison of the used data sources has some limitations, because the reanalysis-based observation data includes climate forcing while the ESM data from piControl runs does not. However, the measured changes to ENSO teleconnection patterns remain weak and are only expected to alter mid 21th century (Yeh et al., 2018).

We find some notable biases in the ENSO-induced climate anomalies among the ESMs. Figure 5 shows the ENSO-90 NBP anomalies from the MLR$_{NBP}$ model using mean values for NBP sensitivity and ENSO-induced climate anomalies from ESM and observations. The strongest biases are in SEA, where the observed ENSO-induced climate anomalies cause almost twice as strong NBP anomalies as the ESM climate. This bias is mostly due to stronger ENSO-induced precipitation anomalies in the observations, especially in MERRA2 (Fig. A3). Other areas with biases are ESAF and SES, where the ENSO-induced





**Figure 4.** ENSO strength and sensitivity of global NBP to ENSO in 17 ESMs, FLUXCOM, CAMS, and TRENDY. The strength of ENSO events is given as the SD of mean annual sea surface temperatures in the Niño3.4 region (x axis). NBP sensitivity to ENSO is the global NBP anomaly to a 1°C anomaly in the Niño3.4 region (y axis). A correlation between ENSO amplitude and NBP sensitivity to ENSO (cor = 0.46) is compensating some of the differences between ESMs.

NBP anomalies are stronger with observed than ESM climate, and SAM and CAF where ESM climate creates stronger NBP anomalies than observed climate.

### 3.4  Biogeochemistry

Differences in biogeochemistry lead to a CV of 56%, which is a 17% increase compared to the differences in the overall ENSO-NBP relationship (Fig. 2 d). This makes biogeochemistry the largest source of uncertainty in the ENSO-NBP relationship.







**Figure 5.** Differences in regional NBP anomalies based on ENSO-induced climate anomalies from ESMs and observations.

Indeed, considering only differences in biogeochemistry produce a larger uncertainty that the apparent uncertainty in the ENSO-NBP relationship. This highlights the compensatory effects that offset some of the differences among the ESMs.

We decomposed the ENSO-90 NBP anomalies into NPP, Rh and fire emissions to reveal what is driving the differences in biogeochemistry (Fig. 6). Global ENSO-induced NPP anomalies are relatively consistent, except for the MIROC ESMs. Fire emissions make up 43% and 32% of ENSO-induced NBP anomalies in E3SM-1-1 and NorCPM1 respectively, while in seven out of the 13 ESMs (with fire emissions) fire explains less than 4% of ENSO-induced NBP anomalies. ENSO-induced fire emissions originate mostly from SEA. Notable deviations among the NBP anomalies are due to uncertainties in the sign of Rh. Rh anomalies can either increase or dampen the effect of reduced NPP. Especially the high NBP anomalies in GFDL-ESM4 are resulting from increased Rh, most of it originating from SEA. However, there is no consistency in the role of Rh for ENSO-induced NBP anomalies. While GFDL-ESM4 and UKESM1-0-LL have comparable NPP anomalies in SEA, NBP anomalies





are 20 times higher in GFDL-ESM4, because the NPP anomalies of UKESM1-0-LL are offset by Rh. This demonstrates that while the NBP sensitivity to climate is suitable to describe $CO_2$ dynamics, it fails to capture the underlying processes.

**Figure 6.** The decomposition of ENSO-induced NBP fluxes (diamonds) into net primary production (NPP), heterotrophic respiration (Rh), and fire, for global fluxes, Southeast Asia (SEA), and N.South-America (NSA). The anomalies represent the carbon fluxes of an ENSO from the 90[th] percentile intensity. Negative values mean reduced NPP and increased Rh and fire emissions.

### 3.5   SEA and NSA

Since the majority of the ENSO-induced NBP anomalies originate from SEA and NSA, we examine the cause of these anomalies in more detail. Figure 7 shows the ENSO-induced precipitation anomalies and the sensitivity of NBP to precipitation

for these two regions. The NBP anomalies in SEA are less constrained than in NSA, both because of the uncertainty in the





ENSO-induced precipitation anomalies and because of the sensitivity of the NBP to precipitation. While the differences in ENSO-induced climate anomalies have a small impact on the uncertainty of the global ENSO-NBP relationship, we find large uncertainties on a regional scale. ENSO-induced precipitation anomalies range between 100 mm yr$^{-1}$ and -210 mm yr$^{-1}$ in SEA and -33 mm yr$^{-1}$ and -180 mm yr$^{-1}$ in NSA.

Although the ENSO-induced NBP anomalies of IPSL-CM6A-LR in SEA reflect the mean NBP anomalies across the other ESMs (Fig. 3), it is resulting from the compensation of two anomalous behaviours. Unlike most ESMs, ENSO creates positive precipitation anomalies in IPSL-CM6A-LR in SEA. However, this atypical behaviour is cancelled out by the negative sensitivity of NBP to precipitation.

**Figure 7.** The composition of ENSO-induced NBP anomalies in S.E.Asia (SEA) and N.South-America (NSA). The x-axis shows the ENSO-induced precipitation anomalies, and the y-axis the sensitivity of NBP to precipitation.





**Table 2.** Differences in the ENSO-induced NBP anomaly pathways in four ESMs. All models have an ENSO-induced NBP anomaly of around -0.98 PgC yr$^{-1}$ per standardised El Niño event. SEA NBP is the ENSO-induced NBP anomaly originating from Southeast Asia (SEA), $T_{frac}$ is the fraction of NBP anomalies in SEA that can be attributed to temperature, $\Delta$ Temp the ENSO-induced temperature anomaly, and $\beta_{NT}$ the sensitivity of NBP to temperature.

| ESM | SEA NBP (%) | $T_{frac}$ (%) | $\Delta$ Temp (°C) | $\beta_{NT}$ (PgC yr$^{-1}$ °C$^{-1}$) |
|---|---|---|---|---|
| ACCESS-ESM1-5 | 9 | 86 | 0.13 | -0.63 |
| CanESM5 | 19 | 105 | 0.07 | -1.62 |
| MPI-ESM1-2-LR | 12 | 74 | 0.32 | -0.41 |
| NorCPM1 | 56 | 0 | 0.03 | -0.02 |

These results highlight the need to describe the complete ENSO-NBP pathway to fully understand the relationship between ENSO and CO$_2$. The pathways describe not the only strength, but rather the "where" and "why" of the ENSO-NBP relationship. We demonstrate the extent of these differences qualitatively by comparing the ENSO-NBP pathways of the four ESMs that are the most similar in terms of their apparent ENSO-NBP strength (Table 2). The ENSO-90 NBP anomalies of these four ESMs fall within a very narrow range of -0.97 to -0.99 PgC, which encompasses the TRENDY anomaly (-0.98 PgC). Within these four ESMs, we focus on the role of SEA, as it is the region with the strongest ENSO-induced NBP fluxes. The share of SEA in the global NBP anomaly is 9% in ACCESS-ESM1-5, 12% in MPI-ESM1-2-LR, 19% in CanESM5, and 56% in NorCPM1. We further use the MLR$_{NBP}$ model to separate the SEA NBP anomalies into the components caused by temperature and precipitation. Temperature anomalies explain 74% and 86% of NBP anomalies in MPI-ESM1-2-LR and ACCESS-ESM1-5 respectively. While the high NBP anomaly in NorCPM1 is exclusively caused by precipitation, the positive precipitation anomaly in CanESM5 even mitigates the overall NBP anomaly caused by temperature. Lastly, we break down whether the temperature-related NBP anomalies are caused by strong ENSO-related temperature anomalies, or by a high sensitivity of NBP to temperature. Although CanESM5 has by far the highest temperature-related NBP anomalies, the actual temperature anomalies are 2 and 4 times smaller than in ACCESS-ESM1-5 and MPI-ESM1-2-LR respectively. The temperature-related NBP anomalies in CanESM5 can be attributed to the high sensitivity of NBP to temperature. The remaining ESMs, ACCESS-ESM1-5 and MPI-ESM1-2-LR, are the most similar in terms of temperature-related NBP anomalies. However, this is because the 2.5 times higher ENSO-induced temperature anomalies in MPI-ESM1-2-LR are somewhat compensated by differences in the sensitivity of NBP to temperature.

## 4 Discussion

We compared the strength and characteristics of the ENSO-NBP relationship in 22 CMIP6 ESMs. The largest source of uncertainty in the simulated ENSO-NBP relationship is due to differences in biogeochemistry. Although differences in ENSO-induced climate anomalies are strong at the regional scale, these errors cancel out globally.





The sensitivity of NBP to Niño3.4 SSTAs is still poorly constrained, with values ranging from -0.13 to -2.00 PgC °C$^{-1}$ SSTA (SD=0.44 PgC °C$^{-1}$ SSTA). Some of this spread can be explained by differences in individual processes like ENSO or the overall sensitivity of carbon fluxes to climatic drivers (Padrón et al., 2022). Although this type of error leads to a large uncertainty in the ENSO-NBP relationship, the errors introduced by a single process do not compromise the consistency of the
results. Differences in ENSO strength, for example, could be balanced out with a single scaling factor.

Another type of error is based on differences in the ENSO-NBP pathways. The combined differences in ENSO-induced climate anomaly patterns and biogeochemistry lead to high uncertainty in the processes behind the ENSO-NBP relationship. Our exemplified description of four ENSO-NBP pathways show that there is little agreement in the origin and drivers of ENSO-induced NBP anomalies, even if they result in similar $CO_2$ growth. The disagreements in the pathways primarily affect the
estimation of regional carbon dynamics. However, the differences in the ENSO-NBP pathways can also distort the prediction of atmospheric $CO_2$ in initialised prediction system. This results from the interaction of initial conditions and the ESM-specific ENSO teleconnection patterns. Although ESMs might have a comparable relationship between ENSO and global NBP, a specific ENSO event can still result in different NBP anomalies. This can be exemplified by the pathways of ACCESS-ESM1-5 and NorCPM1 (Table 2). While both ESMs have a similar average response to ENSO, the NBP anomaly is almost exclusively
from SEA in NorCPM1, while SEA does not play a large role in ACCESS-ESM1-5. Consequentially, the initial conditions and large-scale weather patterns that influence SEA will interact with the ENSO-induce climate anomalies in SEA and co-determine the global NBP anomalies. An initial positive water storage anomaly in NorCMP1, for example, can mitigate the impact of reduced precipitation in SEA and limit the reduction of global NBP.

The main challenge in improving the representation of the ENSO-NBP relationship is to address the errors in biogeochem-
istry. The differences in the sensitivity of NBP to climate are driving the uncertainty in the overall ENSO-NBP relationship. A large portion of this uncertainty can be attributed to the partitioning of NBP. While there is some deviation in the climate sensitivity of NPP, there is no consensus on the sign of Rh sensitivity. Under normal circumstances, NPP and Rh are positively correlated (Baldocchi et al., 2018), because similar climatic conditions favour both fluxes and because the organic material required for Rh is provided through NPP. An exception to this behaviour was proposed by Zeng et al. (2005) which results
from a "conspiracy" between ENSO-induced climate anomalies and plant/soil physiology. The increased temperatures and reduced precipitation during El Niño in the tropics can limit NPP while enhancing Rh. However, we only find this additive effect of NPP and Rh in SEA in GFDL-ESM4 and EC-Earth3-CC. For most other ESMs the ENSO-induced reduction of NPP is partially compensated by the negative Rh anomaly.

A smaller but still significant portion of the uncertainty can be attributed to ENSO-induced climate anomalies. While the
extent of global climate anomalies does not vary much among the ESMs, there is high uncertainty in the spatial distribution of the ENSO-induced anomaly patterns. Although ENSO-induced climate anomalies affect several regions across the globe, they induce the strongest NBP anomalies in SEA and NSA. Important steps in model development would be to reflect the observed balance of ENSO-induced climate anomalies for these regions, and to reduce the strong bias in the climate anomalies in SEA.

The uncertainty among observed carbon fluxes is mostly due to the low interannual variability of NBP in FLUXCOM.
Although FLUXCOM data is not suitable to estimate the absolute ENSO-induced NBP anomalies, it can still be used to assess





the relative contribution of the individual regions to the global ENSO-NBP signal. The ENSO-induced NBP anomalies of CAMS are also below the mean ESM values. This generally lower response of land carbon fluxes to ENSO in inversion data compared with ESM has been reported by others (Bastos et al., 2018, 2020). Although there are differences between inversion products based on atmospheric transport models, assimilated observations, prior carbon fluxes and fossil fuel emission data

(Gaubert et al., 2019), the differences among inversion products are small compared to the differences among ESM (Bastos et al., 2020).

## 5   Conclusions

Although ESMs are able to reproduce the relationship between ENSO SSTA and $CO_2$, there is little agreement in the processes behind this relationship. This is because uncertainties on the regional scales are balanced out when fluxes are aggregated

globally. Consequentially, the correct reproduction of atmospheric $CO_2$ variability in assimilation runs forced by SSTAs does not necessarily mean a good representation of atmospheric circulation patterns and biogeochemistry. It could also result from an alternative ENSO-NBP pathway that does not reflect observable processes. We attribute this high uncertainty in the ENSO-NBP relationship to our limited understanding of the sensitivity of terrestrial carbon fluxes to climate. This ongoing challenge is due to the low availability and quality of carbon flux observations and limits the ability of ESMs to reproduce the interannual

variability of the terrestrial carbon cycle. This is, however, where the ENSO-NBP relationship provides an untapped potential. Instead of tuning ESMs with local observations of carbon flux data, the models could be optimised to reproduce the ENSO-induced, large-scale patterns. This can be a favourable alternative to the traditional approach because of the availability of high-accuracy data from atmospheric $CO_2$ measurements and continental carbon flux anomalies from inversion products. Improving the reproduction of the regional response of terrestrial carbon fluxes to ENSO-induced climate anomalies is not

only a tangible goal but can also lead to ESMs with a better ability to simulate the interannual variability of global carbon fluxes and an improved predictability of the global carbon cycle.

*Code and data availability.*   The data and code to produce the figures shown in this study are available at [TBD]





## Appendix A

**Figure A1.** Global annual NBP anomalies against Niño3.4 SSTA in observational and ESM data. The red regression lines are fitted to SSTA values > 0 and the blue lines to values < 0, the grey area represents the 95% confidence interval of the regression lines. The grey vertical lines are the 10th and 90th percentile of ENSO SSTAs.







**Figure A2.** Regional ENSO-induced annual temperature anomalies. The values are the sensitivity of annual temperature anomalies to Niño3.4 SSTA ($\beta_{ET}$).

*Author contributions.* The Study was conceptualised by ID, AB and TI, ID developed the methodology, performed the analysis, and wrote
the original draft, AB and TI reviewed and edited the manuscript.

*Competing interests.* The authors declare that they have no conflict of interest.



**Figure A3.** Regional ENSO-induced annual precipitation anomalies. The values show the sensitivity of annual precipitation anomalies to Niño3.4 SSTA ($\beta_{EP}$).

*Acknowledgements.* This project has received funding from the European Union's Horizon 2020 research and innovation programme (4C project, grant agreement No 821003). The authors thank David Nielsen for reviewing the manuscript.





**Figure A4.** The effect of differences in ENSO-induced climate anomaly patterns on regional NBP anomalies. The climate anomaly patterns of a 90$^{\text{th}}$ percentile El Niño from ESMs and climate reanalysis products are applied to a linear regression model to reproduce global NBP anomalies.

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
