# Peer review of "Compensatory effects conceal large uncertainties in the modelled processes behind the ENSO-CO2 relationship"

_Earth System Dynamics, 2024_

## Author Comment (AC1)

**Author answers to the comments of the manuscript "Compensatory effects conceal large uncertainties in the modelled processes behind the ENSO-CO₂ relationship"**

**RC1:**

Review of "compensatory effects conceal large uncertainties in the modelled processes behind the ENSO-CO2 relationship", by Dunkl et al.

This is a well written and explained paper exploring in some depth how the land carbon cycle responds to El Nino events and getting beneath the skin of multiple ESMs. The study breaks apart the response into the magnitude of ENSO, the spatial patterns of climate teleconnections and then the land-carbon sensitivity to these. The latter controls the majority of the spread in modelled responses.

The paper is useful both as a process study on drivers of carbon cycle variability but also in terms of guiding model groups/development plans and evaluation techniques. I recommend publication with minor revisions.

> We thank Chris Jones for the careful reading of the manuscript. His summary reflects our views on the matter well, and we are grateful for the constructive comments.

I have one major concern though, and that is the use of TRENDY model results as "observations" against which to evaluate ESMs. This is problematic in a couple of ways

- Firstly these are clearly not "observations" – they have some link to observed meteorology as they are driven by it – but the response is very much a model response

- Secondly, and maybe more important – they are not at all independent of the models you are evaluating. There is a very big overlap between the land schemes in CMIP6 ESMs and the land models used for TRENDY.

So I am afraid you simply cannot use these in the way you do now as observations.

> We agree that the TRENDY data does not qualify as "observations". The TRENDY ensemble is based on the output of land surface models, some of which are components of the ESMs used in this study.

I think this issue has a couple of solutions – depending on your appetite for further study. The simple solution is to drop TRENDY models. You have two other "observation" data sets (which are also not pure obs – as per first objection above – but they are closer to this and they are independent of CMIP6 land schemes). The paper could stand equally well using these two datasets and I don't think the conclusions would be affected.

A more thorough, and satisfying, outcome could be to make use of the overlap and to see TRENDY results as part-way between the CMIP6 ESMs and the observations. You could even explore a pair-wise comparison for many of the TRENDY/CMIP shared land models (e.g. compare UKESM with JULES, or MPIESM with JSBACH). Where would individual TRENDY models sit on figure 4 for example? I assume they would all be at the same x-axis location (as Nino3.4 is imposed on them), but they would span the same vertical extent as CMIP models?? Can we learn anything by comparing the offline and coupled land schemes. I think in general treating TRENDY as models rather than obs is a better way forward.

> We will describe the constraints of using TRENDY data as a comparison in the methodology. In the results section, we will separate out the TRENDY data from CAMS and FLUXCOM and make a clear distinction. The TRENDY data will be shown, but it will not be part of the "Observations" in figures 2 and 5.

Other, minor comments

-        I miss any mention of other studies which have tried to constrain CMIP outcomes based on interannual variability. Jones et al (2001) did an early exploration of how a single ESM responded to ENSO and since then several studies have used this as a constrain on future behaviour – notably Cox et al (2013). Do your results have any implications for this approach?

> Cox et al 2013 provides a good foundation on the scale of the uncertainty in the sensitivity of tropical carbon fluxes to temperature anomalies in CMIP5 models, and how this uncertainty relates to atmospheric CO2 IAV. We will link our findings on CMIP6 models regarding this issue and add the role of spatial heterogeneity in the sensitivity of tropical carbon fluxes to climate anomalies.

-        Re CMIP6 model selection – I would recommend caution when using multiple models which are very close variants – e.g. NoESM2-LM and -MM are essentially the same model except for spatial resolution. The land surface is identical. Likewise the various CMCC variants. Do they really add extra info to this particular study of land response? (maybe they do if the ENSO characteristics differ for example). It might simplify things to reduce the sampling to only one variant from each model family. This might feel like you are taking a smaller sample, but actually by double-sampling the same model you may skew the results.

> Although we do not expect strong resolution dependent effects from the land surface models, ENSO dynamics can be altered by a higher resolution of the ocean or atmosphere. Although the two CMCC variants have similar components we still see strong differences in ENSO strength for example.

-        It is often quoted that a multi-model mean performs better than individual members (see Jones et al 2023 for a discussion on this for CMIP6 carbon cycle at regional scale). It would be interesting to see the CMIP6 multi-model mean in your evaluation as well as single models.

> The multi-model mean can provide an interesting additional insight. We will add it to the results where feasible.

- Figure 3 – can you zoom in on the panels? It is very hard to read much into the results for regions other than SEA and NSA. I realise this would break the nice feature of having the same x-axis for all panels, but I think the other panels are just too small to see much clearly.

> We will create a new figure with one x-axis range for the first two panels, and another range for the remaining panels.

- The inverse relationship between Nino magnitude and NBP sensitivity is interesting – can you comment why you think this might come about? I cannot think of a process-reason for it – why would models with bigger ENSO have lower sensitivity? Is this an artefact of trying to cancel out errors in a model calibration stage? It would be interesting if all model groups had done that!

> We also suspected the relationship to arise from model tuning. However, different model developers could not confirm this in personal communication. Can can still add this possible explanation to the manuscript.

- I like that you split into NPP and respiration – that's nice (also seen in Jones et al 2001). Did you think about any obs for this step? I know MODIS NPP is not perfect, but could be useful to identify spatial patterns of NPP for example even if the absolute magnitude is not reliable.

> Thank you for the suggestion. It would be beneficial to have this division of NBP in the observations as well. However we decided against using different data sources for the individual carbon fluxes as this would bring in too many constrains.

- A final comment – you discuss a lot, and very well, the differences between models and how the two ends of the responses differ. But actually I am also struck that generally most models do OK. For example my first reaction on seeing figure 5 is that generally ESM vs OBS picks up very good extent of the signal between regions. I think it would be useful to say this – that actually CMIP models are not bad. OK they differ in details, and some can be far away from the obs for some metrics. But overall the agreement is encouraging.

> This feeds into the earlier comment on the multi-model mean. Indeed figure 5 visualizes the overall agreement in the spatial patterns quite well. We will add some words on this overall agreement to balance the view.

- Jones 2001: https://journals.ametsoc.org/view/journals/clim/14/21/1520-0442_2001_014_4113_tccrte_2.0.co_2.xml
- Cox 2013: https://www.nature.com/articles/nature11882
- Jones 2023: https://agupubs.onlinelibrary.wiley.com/doi/full/10.1029/2023AV001024

**Citation**: https://doi.org/10.5194/esd-2024-7-RC1

---

## Author Comment (AC2)

**Author answers to the comments of the manuscript "Compensatory effects conceal large uncertainties in the modelled processes behind the ENSO-CO$_2$ relationship"**

**RC2:**

Comments on "Compensatory effects conceal large uncertainties in the modelled processes behind the ENSO-CO2 relationship"

This manuscript investigates ENSO-CO2 relationship in 22 CMIP6 ESMs by describing their characteristics ENSO-NBP pathways, and explain processes which contribute most to the overall uncertainties in ENSO-CO2 relationships among ESM. And authors find that the largest source of uncertainty is the differences in the sensitivity of NBP to climate. Overall, the manuscript is concise and clear. Here are some minor suggestions.

> We thank the reviewer for the careful reading of the manuscript and their constructive comments.

(1) In abstract: "look beyond the top-down ENSO-CO2 relationship in 22 CMIP6 ESMs", how to understand the "top-down"

> We will rephrase this to provide a better description of our approach to disentangle the ENSO-CO2 relationship.

(2) Page2Line30: "Tropical carbon flux anomalies lag behind ENSO by three to six months (Zhu et al., 2017)", this another paper may be a good reference here which calculate the lead-lag between ENSO and CGR/NBP. "Wang, J., Zeng, N., & Wang, M. (2016). Interannual variability of the atmospheric CO2 growth rate: roles of precipitation and temperature. Biogeosciences, 13(8), 2339-2352."

> The two publications provide comparable results in terms if lag time. We decided to cite Zhu et al because it provides the spatial patterns of the lag. However, we see the benefit of adding another citation that calculates the lag to NBP instead of GPP.

(3) Page 4, which periods do you use for reanalysis products?

> We used the whole period of available data for each source at the time of the analysis. We will add the specific range of years to the methods section on page 4.

(4) Page 9, Line 170-171, you may calculate and show the spreads in ENSO-induced temperature and precipitation for each region in the plot.

> We will add a top row to these plots similar to Figure 2.

(5) Page11Line 197-198, "the high NBP anomalies in GFDL-ESM4 are resulting from increased Rh", increased Rh => reduced Rh? In Figure 6, NBP anomalies in MIROC-ESMs are nearly totally caused by Rh. Maybe need to mention it in the text.

> In the global panel of Figure 6 we see that the diamond representing the NBP anomaly aligns with the sum of NPP, Rh and Fire. The NBP anomaly is larger than all of the single fluxes. This means that a low NPP anomaly (plants take up less carbon) meets a high Rh anomaly (ecosystems respire more carbon). We use this alternative sign of carbon flux anomalies to visualize the composition of NBP anomalies, and describe the direction of the fluxes in the figure caption.

**Citation**: https://doi.org/10.5194/esd-2024-7-RC2